# Bulgarian Experience in Vaginal Electronic Brachytherapy for Gynecologic Cancers’ Treatment—First Results

**DOI:** 10.3390/jcm13247849

**Published:** 2024-12-23

**Authors:** Virginia Payakova, Angel Yordanov, Desislava Kostova-Lefterova, Nikolay Mutkurov, Ilko Iliev, Marin Valkov, Elitsa Encheva, Desislava Hitova-Topkarova

**Affiliations:** 1Scientific and Innovative Program Med for Health, Medical University Pleven, 1, Saint Kliment Ohridski Street, 5800 Pleven, Bulgaria; vpayakova@gmail.com (V.P.); angel.jordanov@gmail.com (A.Y.); dessi.zvkl@gmail.com (D.K.-L.); ilkoiliev_92@abv.bg (I.I.); marinvalkov@abv.bg (M.V.); 2Department of Radiotherapy, UMHAT “Dr. Georgi Stranski”, 8A Georgi Kochev Blvd., 5809 Pleven, Bulgaria; 3Department of Gynecologic Oncology, UMHAT “Dr. Georgi Stranski”, 8A Georgi Kochev Blvd., 5809 Pleven, Bulgaria; 4National Cardiology Hospital, 65 Konyovitsa Street, 1309 Sofia, Bulgaria; 5Complex Oncology Centre, 86 Demokratsia Blvd., 8000 Burgas, Bulgaria; n_mut@abv.bg; 6Department of Radiotherapy, UMHAT “Saint Marina, 1 Hristo Smirnenski Blvd., 9010 Varna, Bulgaria; dr.encheva@gmail.com; 7Faculty of Medicine, Medical University Varna, 55, “Professor Marin Drinov” Street, 9002 Varna, Bulgaria

**Keywords:** endometrial cancer, cervical cancer, electronic brachytherapy

## Abstract

**Background/Objectives**: The objective of this study is to prospectively collect dosimetric and clinical data on vaginal cuff electronic brachytherapy and propose a protocol for the procedure. **Methods:** Twenty-five patients who had proven endometrial or cervical carcinoma and had undergone radical hysterectomy have been treated with vaginal cuff electronic brachytherapy. Treatment session durations and doses to the targets and the organs at risk have been extracted from the treatment planning software. Patients have been followed up for early side effects for 3 months. **Results:** Treatment session times ranged from 3.0 to 6.6 min. Mean coverage of the planned treatment volume with 100% of the prescribed dose was 90%, and with 95% of the prescribed dose was 95%. Doses in the bladder were lower than those achieved in previously published studies with a mean D2cc of 4.7 Gy, and doses in the rectum were higher with a mean D2cc of 5.3 Gy. The first-month adverse events included eight G1 and three G2 toxicities, while the events registered on the third month were two G2 vaginal dryness events and one G1 urinary tract obstruction, of which only one patient with vaginal dryness did not respond to local treatment. No local relapses have been detected. **Conclusions**: Vaginal cuff electronic brachytherapy has demonstrated safety and effectiveness.

## 1. Introduction

In 2022, more than one million new cases of cancers to the uterine body and cervix were diagnosed with the following distribution: 14.1 per 100,000 for cervix uteri and 8.4 per 100,000 for corpus uteri, accounting for 11.2% of all women’s malignancies worldwide [1]. The treatment of gynecologic cancers is complex and depends on multiple prognostic factors. Modalities such as surgery, chemotherapy, external beam radiotherapy (EBRT), and brachytherapy (BT) can be recommended alone or combined. Guidelines by the National Comprehensive Cancer Network and the European Society of Therapeutic Radiation Oncology [2,3,4] for management of endometrial and cervical cancer define the role of radiotherapy. The primary goal of radiotherapy is to deliver a precise amount of a high dose of ionizing radiation to a specific region of interest to ensure the death of malignant cells or to slow down their growth by destroying their DNA while harming as few healthy cells as possible in the surrounding healthy tissue [5]. BT is a type of radiation therapy that consists of placing radioactive sources inside or in near proximity to a tumor, most often in the treatment of cervical, prostate, breast, and skin cancer. This modality has been used since the discovery of radioactivity and allows an optimal dose distribution, enabling a higher dose of radiation in a short period of time directly to the target area with minimal exposure to surrounding tissues [6]. The main difference between the radionuclide BT and electronic BT (EBT) is the nature of the source: a radioactive isotope, or a low-energy X-ray generator, respectively. In 1999, a publication by David J. Brenner et al. discussed the clinically-related factors and the biological effectiveness of low-energy X-rays emitted by miniature X-ray devices [7]. The authors concluded that the use of lower energies reduces the requirements for shielding of the surrounding environment and increases the radiobiological effectiveness. The lack of radioactive material can eliminate radioactive waste, as well as most radiation incidents. Considering the low energy of the source, the dose fall-off is rapid and results in lower doses in surrounding tissues [8].

The purpose of the current study is to develop a protocol for vaginal cuff electronic brachytherapy treatment of gynecological cancer and to present the first clinical results.

## 2. Materials and Methods

Twenty-five patients were treated with vaginal cuff EBT. Analysis of the dosimetry data and the early toxicity in the first 3 months after treatment was performed. Patient selection criteria included histologically proven endometrial cancer or cervical cancer after radical hysterectomy (RH), with or without pelvic lymph node dissection, and who were generally subject to adjuvant BT alone or after external beam radiotherapy (EBRT). All of them were over the age of 18 with signed informed consent forms and a lack of general contraindications for BT. Patients were scheduled for treatment after surgical wounds had completely healed and the vaginal mucosa was confirmed to be intact by gynecological examination. In case of indications for EBRT before EBT, image-guided volumetric modulated arc therapy was delivered with a daily dose of 1.8–2 Gy to a total dose of 45–50.4 Gy. EBT treatment was performed on a mobile system Xoft^®^ Axxent^®^ Electronic Brachytherapy (eBx^®^) System^®^ (Elekta, Stockholm, Sweden) with a low-energy miniature X-ray source 50 kV (Figure 1). The X-ray source is integrated with a cooling sheath into a multi-lumen catheter, delivering high-dose rate, low-energy radiation [9].

To irradiate the vaginal cuff with precisely defined dose distribution, vaginal applicators specially designed for that system were used. Applicators are cylindrical in shape, made of medical-grade polymers, and provide transmission characteristics specifically for the low-energy X-rays emitted by the EBT source [10]. The available cylinders are four different sizes in diameter: 20 mm, 25 mm, 30 mm, and 35 mm, with an external length of 100 mm. (Figure 2). The selection of the appropriate diameter and the estimation of the total vaginal length were performed after a gynecological examination shortly before the treatment.

Patient preparation for computed tomography (CT) simulation began with patient positioning over a vacuum bag and the base plate and clamp. The bladder was filled with 100 mL of saline through a Foley catheter, and the applicator was inserted while the patient was in a lithotomy position. The applicator was connected and locked to the clamp according to the manufacturer’s recommendations, and the vacuum bag was shaped around the patient’s body after the legs were rested straight to avoid changes in the position of the applicator. A CT of the pelvic area of interest was acquired with a slice thickness of 2 mm, and the images were checked visually for air gaps or unfavorable positioning of the applicator. In case of presented air gaps or displacement of the applicator, corrective actions were taken immediately. The CT images were transferred to the treatment planning software (TPS), BrachyCare version 1.1.0, based on task group no. 43 report: a revised American Association of Physics in Medicine (AAPM) protocol for brachytherapy dose calculations (TG-43). The defined organs at risk (OAR) were bladder, rectum, and sigmoid colon. The planning target volume (PTV) was defined as the upper third of the vagina +5 mm outside of the applicator to ensure coverage of the vaginal mucosa. The prescribed physical dose per fraction varied between 5 to 6 Gy in patients previously treated with EBRT, and 7 Gy when only EBT was applied. Dosimetry planning included the following steps: defining dose per fraction, dose prescription points, the implant geometry, the length of the channel with irradiation positions (the active length), and optimizing the source dwell time distribution along it. Validation of the plan was a joint decision between the team members, including a medical physicist and a radiation oncologist, in terms of achieving target coverage and constraints to OARs. The general approval criteria are presented in Table 1.

The treatment plan was transferred to the Xoft Axxent controller system, and the treatment procedure followed the manufacturer’s recommendations for quality assurance, including source length verification, position verification of the catheter, temperature and pressure corrections, cooling system function checks, and source strength calibration. The printed data from the TPS were double checked for consistency with the data imported in the controller.

The described process was repeated for each treatment session to a total of 2 to 3 fractions performed over 2 to 3 weeks. The total physical dose delivered by EBT to the vaginal cuff was between 10 and 21 Gy. Patient follow-up was scheduled on the 1st, 3rd, 6th, 9th, 12th, 18th, 24th, and 36th months after the treatment by gynecological examination. PET/CT was conducted three months after EBT treatment and scheduled annually for 5 years. Toxicity was evaluated by the radiation oncologist and the gynecologist on each visit and scored using Common Terminology Criteria for Adverse Events (CTCAE) version 5. During the follow-up period, patients were encouraged to report any adverse events.

## 3. Results

All 25 patients were treated successfully in accordance with the planned schedule. Their median age was 58 years (range 35–82).

No events of machine malfunction occurred, and the steps of the treatment protocol were strictly followed. Overall, six patients received EBRT and two fractions of EBT to the upper third of the vaginal cuff. Of them, three patients had an incidental finding of cervical carcinoma- one stage IA2 and two stage IB1. The histologies were adenocarcinoma G2 in one case and non-keratinizing squamous cell carcinoma G2 in two cases. The other three patients in the group were diagnosed with endometrial cancer with the following histologies and stages: endometrioid adenocarcinoma G1 stage II, serous carcinoma stage IA, and serous carcinoma IB. Cylinder diameters (d), active lengths, and the average value for the treatment time (*t*) per fraction are presented in Table 2. The total physical dose was 12 Gy for patients 1 to 5. Patient 6 was prescribed a dose of 10 Gy as a result of having received the highest EBRT dose (50.4 Gy) and the necessity for the widest applicator, which would result in compromising the OAR tolerance doses.

Nineteen patients were treated with 7 Gy per fraction to a total physical dose of 21 Gy. All of them were diagnosed with endometrial cancer with the following histology distribution: 14 endometrioid adenocarcinomas G2, two G1, and two G3. Thirteen patients were stage IA and five were stage IB. One patient had serous carcinoma stage II and received only vaginal cuff EBT due to contraindications for pelvic EBRT. Data for cylinder sizes (d), active lengths, and the average value of treatment time per fraction for the group of patients when only EBT is applied are presented in Table 3.

PTV coverage was individually assessed for each fraction of treatment in accordance with the clinical situation. The mean values of PTV coverage and achieved dose distribution in OARs are presented in Table 4 and Table 5. Figure 3 represents a typical dose distribution case in the sagittal plane.

In the group of patients who received EBRT before EBT, the cumulative maximal doses in 2 cm^3^ of the contoured organ volume were calculated in equivalent dose in 2 Gy fractions (EQD2), assuming the location of the volume was the same. However, the sigmoid was not contoured for the EBRT plans in some patients as they had been treated at other departments and the sigmoid was not included in their contouring protocols. The results from the calculation for each patient are presented in Table 6.

All 25 patients were followed up for early side reactions in the first 3 months after treatment. There were no abnormal findings in blood analysis or biochemical parameters. No disease recurrences were found on PET/CT or contrast-enhanced CT and gynecological examinations. Overall, eight G1 and three G2 toxicities were registered in the first month after treatment and resolved by the end of the third month. For 3-month follow-up, only one event G1 and two G2 were registered among all irradiated patients.

## 4. Discussion

EBT for gynecological cancers is an easy to implement modality in the daily practice of the radiotherapy department. The mobility of the system equipment and minimal shielding requirements [11] allowed for flexibility in choosing between treatment rooms according to their availability. Treatment with a vaginal applicator was performed in an outpatient setting, and no hospital stay was required. The most frequently used applicator sizes were 25 and 30 mm diameter, followed by 35 mm, while 20 mm was never used. Treatment times were in the range of 3 to 6.6 min, depending on the size of the PTV as well as on individual dwell position times calculated during optimization depending on the proximity of OARs. The analysis of the results from the current study showed that the values for V50% and V35% for bladder are lower compared to other published studies [12,13,14,15]. The results for the D2cc, D1cc, and D0,1cc values for bladder are comparable with the published data by Lozares-Cordero [16]. The D2cc value for rectum from the current study is higher than the reported value by Lozares-Cordero et al., while V35 and V50 for rectum are higher or equal. A possible explanation for the higher reported rectal doses is in the overlapping of PTV with the contour of the critical organs, especially in the case of bigger applicators. Measuring the thickness of vaginal mucosa using ultrasound [17] or evaluation of the vaginal cuff using magnetic resonance imaging (MRI) [18] are methods that can be implemented to define the depth of the PTV individually and reduce unnecessary doses to the rectum while respecting the depth of the vaginal lymphatics [19]. Reduction in the irradiated length of the vagina can also produce smaller PTVs and has shown adequate local control [20], while in this study the vaginal length was measured and the upper third was treated. Another possible reason for this trend of higher rectal doses could be connected to contouring uncertainties when defining the exact transition from rectum to sigmoid colon depending on patient-related factors—postsurgical changes in pelvic anatomy and individual functional variations. All patients are simulated on the available CT equipment with a 70 cm gantry and treated with legs resting straight on the table due to patient immobilization and transportation issues, thus leading to a closer position of the rectum to the PTV and higher rectal doses. According to Roel et al. [21], the addition of a knee support device can shift the rectum dorsally and push rectal filling caudally. The usage of knee support can be an improvement in the current immobilization protocol, especially when a patient is treated lying on the CT table. The possible reason for the lower bladder doses is in the followed patient preparation and treatment protocol, including 100 mL saline filling of the bladder. The higher rectal doses did not lead to severe early adverse events. One patient presented with abdominal pain G1 and another presented with both abdominal pain G1 and diarrhea G1 in the first month after treatment. Both were resolved by the third month by oral administration of probiotics. There were two cases of vaginal discharge G1, two vaginal necrosis G1, one urinary incontinence G1, two vaginal necrosis G2, and one vaginal dryness G2. Patients with G2 toxicities were treated locally with hyaluronic acid ovulae. Only one G2 vaginal dryness persisted after the third month in the patient who had been treated with 50,4 Gy EBRT and 10 Gy EBT. The other third-month events included a urinary tract obstruction G1 and a vaginal dryness G2, which were treated successfully by spasmolytics and hyaluronic acid, respectively. The success of managing vaginal toxicities is similar to a previously published study by Markowska et al. [22].

## 5. Conclusions

To our knowledge, this is the first study assessing early toxicity after vaginal EBT according to CTCAE version 5. Previous studies are focused on the use of Radiation Therapy Oncology Group (RTOG) and CTCAE version 3 and 4 criteria. While long-term follow-up is still undergoing, our first experience with vaginal cuff EBT proves to be safe and effective, with the potential to provide similar clinical results to conventional brachytherapy when the suggested protocol is followed. Further evaluation of local control, late toxicity, and quality of life will be of great value in defining the benefits of vaginal cuff EBT.

## Figures and Tables

**Figure 1 jcm-13-07849-f001:**
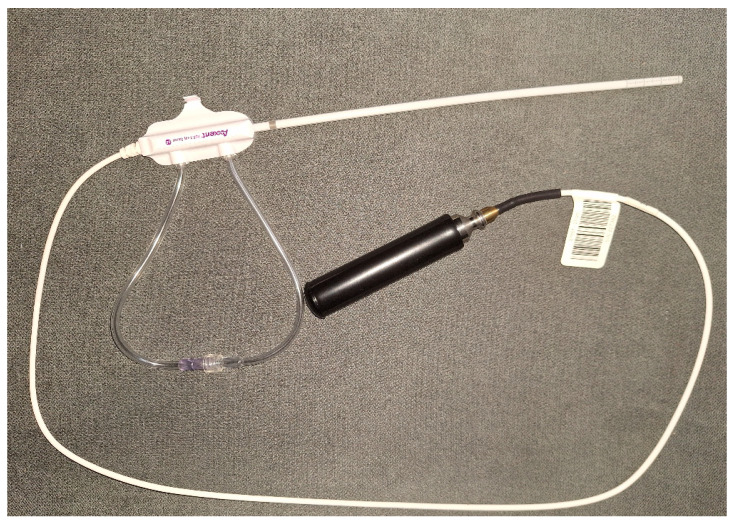
Low-energy X-ray source with a miniature X-ray tube integrated with a cooling sheath into a multi-lumen catheter.

**Figure 2 jcm-13-07849-f002:**
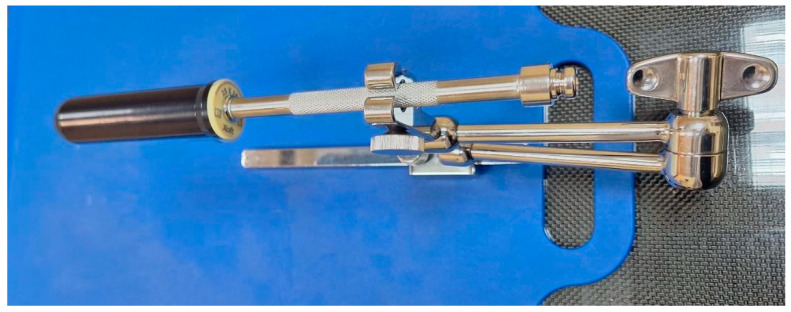
Vaginal applicator attached to the base plate and clamp.

**Figure 3 jcm-13-07849-f003:**
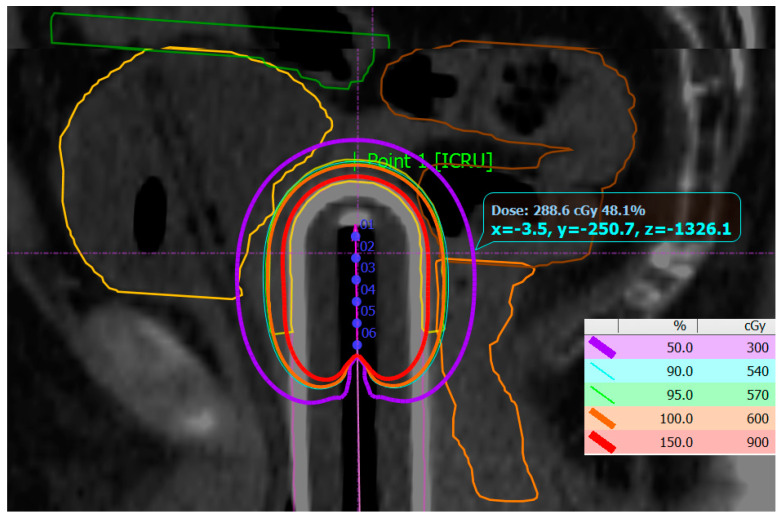
Dose distribution from 150% of the prescribed dose (red isocurve) to 50% (purple isocurve). The PTV is generated 5 mm from the applicator surface (yellow) and overlaps with 100% (orange) and 95% of the prescribed dose (green). The dwell positions among the active length of the applicator channel are numbered 1 to 6. The OARs are bladder (yellow), sigmoid (brown), and rectum (orange), as well as the most proximal small bowel loop (green).

**Table 1 jcm-13-07849-t001:** Treatment plan goals and constraints per session. V100 is the volume of PTV receiving 100% of the prescribed dose. D_2cc_ is the maximum dose to 2 cm^3^ of the OAR.

V100	Bladder D_2cc_	Rectum D_2cc_	Sigmoid D_2cc_
>90%	<7.3 Gy	<5.3 Gy	<6 Gy

**Table 2 jcm-13-07849-t002:** Data for the group of patients after EBRT.

Patient Number	d (mm)	Active Length (mm)	*t* (min)
1	30	30	3.6
2	30	25	3.8
3	25	25	2.7
4	35	25	3.8
5	30	30	4.3
6	35	25	3.8

**Table 3 jcm-13-07849-t003:** Data for the group of patients treated only with EBT.

Patient Number	d (mm)	Active Length (mm)	*t* (min)
1	25	30	3.1
2	30	30	5.3
3	35	30	5.8
4	30	30	4.7
5	25	25	3.2
6	25	25	3.1
7	30	25	4.2
8	35	30	5.9
9	35	25	5.4
10	25	25	3.0
11	25	25	3.2
12	25	25	3.2
13	30	30	4.7
14	25	25	3.0
15	25	25	3.1
16	30	30	4.6
17	30	30	4.7
18	35	35	6.6
19	30	30	4.2

**Table 4 jcm-13-07849-t004:** V90, V95, V100, and V150, the percentage of the PTV receiving 90%, 95%, 100%, and 150% of the prescribed dose; SD, standard deviation.

Data for the Group of Patients After EBRT			
	Mean Value (%) and SD (%)	Maximal Value (%)	Minimal Value (%)
PTV V90	97.9% ± 0.96%	99.5%	96.4%
PTV V95	93.9% ± 1.78%	97.2%	91%
PTV V100	86.3% ± 6.98%	92.1%	84%
PTV V150	25.2% ± 5.57%	36.5%	17.1%
Data for the group of patients when only EBT is applied			
PTV V90	98.6% ± 0.91%	99.6%	97.4%
PTV V95	96.0% ± 2.4%	98.3%	93.6%
PTV V100	91.9% ± 4.02%	96.3%	86.1%
PTV V150	29.9% ± 6.45%	34.9%	18.9%

**Table 5 jcm-13-07849-t005:** The mean achieved dose distribution in the OARs, where V50% and V35% are the volumes receiving 50% or 35% of the prescribed dose, and D_2cc_, D_1cc_, and D_0,1cc_ are the maximum doses in the following contoured organ volumes: 2 cm^3^, 1 cm^3^, and 0.1cm^3^. SD, standard deviation.

Data for the Group of Patients After EBRT					
	V50% (%) and SD (%)	V35% (%) and SD (%)	D_2cc_ (Gy) and SD (Gy)	D_1cc_ (Gy) and SD (Gy)	D_0,1cc_ (Gy) and SD (Gy)
Bladder	6.3% ± 3.4%	12.8% ± 5.5%	4.7 Gy ± 1.2 Gy	5.4 Gy ± 1.2 Gy	6.9 Gy ± 1.4 Gy
Rectum	24.6% ± 13.9%	37.8% ± 16.9%	5.4 Gy ± 1.4 Gy	6.3 Gy ± 1.4 Gy	8.1 Gy ± 1.7 Gy
Sigmoid	4% ± 3.3%	9.1% ± 8.0%	2.1 Gy ± 0.7 Gy	2.7 Gy ± 0.9 Gy	4.4 Gy ± 1.8 Gy
Data for the group of patients when only EBT is applied					
Bladder	4.1% ± 2.2%	9% ± 4.1%	4.7 Gy ± 0.95 Gy	5.3 Gy ± 1.0 Gy	6.7 Gy ± 1.3 Gy
Rectum	19.4% ± 12.3%	32.9% ± 17.5%	5.4 Gy ± 1.3 Gy	6.3 Gy ± 1.4 Gy	8.3 Gy ± 1.6 Gy
Sigmoid	3.3% ± 5.3%	7.1% ± 9.4%	2.7 Gy ± 1.4 Gy	3.3 Gy ± 1.6 Gy	4.8 Gy ± 2.2 Gy

**Table 6 jcm-13-07849-t006:** Cumulative doses in EQD_2_ for the OARs in patients who received EBRT. The α/β ratio is 3.

Patient Number	Bladder D_2cc_ EQD_2_ (Gy)	Rectum D_2cc_ EQD_2_ (Gy)
1	70.55 Gy	59.61 Gy
2	60.11 Gy	84.69 Gy
3	53.50 Gy	52.23 Gy
4	66.02 Gy	57.90 Gy
5	68.24 Gy	66.03 Gy
6	55.80 Gy	61.12 Gy

## Data Availability

All available data can be provided by the corresponding author on demand.

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
