# Peer review of "Bulgarian Experience in Vaginal Electronic Brachytherapy for Gynecologic Cancers’ Treatment—First Results"

_jcm, 2024, doi:10.3390/jcm13247849_

Round 1

Reviewer 1 Report

Comments and Suggestions for Authors

The study explores the use of vaginal electronic brachytherapy (EBT) as an adjuvant treatment for patients with endometrial or cervical cancer. It focuses on the protocol development and clinical outcomes for patients who underwent vaginal cuff EBT following radical hysterectomy. The research involves the treatment of 25 patients and evaluates treatment dosimetry, session duration, and early toxicity, assessing both the effectiveness and safety of the approach. this study provides promising insights into vaginal cuff EBT, highlighting its safety and effectiveness with a new protocol, although longer-term data and further optimization could enhance its application.

I have the following concerns:

1)        The authors should include the characteristics of the 25 patients, their tumor stage, and their type of cancer, whether endometrial or cervical. Is it not clear how many patients with endometrial cancers and how many patients had cervical cancers

2)        They should also mention how many of them received EBRT + BT or only BT

3)        Did all the patients treated with brachytherapy alone receive the total three fractions with 7 Gy, or should they reduce the dose due to rectal and sigmoid exposure?

4)        In the combination arm (EBRT + BT), it will be interesting for the readers to know the whole dose exposure to the OAR (rectum, bladder, sigmoid) after calculating both doses like in Embrace studies protocols. The authors should mention that in their results

5)   In Tables 3 and 4, the authors should stress the need for a clear presentation of the data. They should mention the unit near the results so that readers who don’t have experience with brachytherapy will understand whether it is % or Gy.

6) Higher Rectal Dose: Compared to other studies, rectal doses were relatively high, which may be attributed to patient positioning and applicator size, potentially increasing the risk of rectal toxicity. they should ecplain that in the discussion.

7) Limited Follow-Up Duration: Early toxicity was only observed over three months; longer follow-up could better assess late toxicity and long-term outcomes.

Comments on the Quality of English Language

should be improved and the gramatic failures should be corrected

Author Response

Thank you very much for taking the time to review this manuscript. Please find the detailed responses below and the corresponding revisions in the revised manuscript.

Comments 1: The authors should include the characteristics of the 25 patients, their tumor stage, and their type of cancer, whether endometrial or cervical. Is it not clear how many patients with endometrial cancers and how many patients had cervical cancers.

Response 1: Thank you for the valuable insight. The characteristics are now descibed in the revised manuscript in rows 150- 167.

Comments 2: They should also mention how many of them received EBRT + BT or only BT

Response 2: The clear statement of the number of patients in both groups is now written in the revised manuscript in row 150 and row 163.

Comments 3: Did all the patients treated with brachytherapy alone receive the total three fractions with 7 Gy, or should they reduce the dose due to rectal and sigmoid exposure?

Response 3: In row 163, where the number of patients in the EBT alone is defined, we have also brought clarity on the prescribed dose and fractionation.

Comments 4:  In the combination arm (EBRT + BT), it will be interesting for the readers to know the whole dose exposure to the OAR (rectum, bladder, sigmoid) after calculating both doses like in Embrace studies protocols. The authors should mention that in their results.

Response 4: Thank you for the suggestion as it brings more depth to the data and is an interesting addition. In the revised manuscript, we have added this information in rows 198- 203, and created an additional Table 6. Another table, which is now numbered as Table 1, was added to define our basic decision making criteria as suggested by another reviewer. Our dose constraints were also based on EMBRACE II suggestions.

Comments 5: In Tables 3 and 4, the authors should stress the need for a clear presentation of the data. They should mention the unit near the results so that readers who don’t have experience with brachytherapy will understand whether it is % or Gy.

Response 5: Due to the addition of a new Table 1, in the revised manuscript these are now Tables 4 and 5. We have considered your suggestion and added the units of measurement.

Comments 6. Higher Rectal Dose: Compared to other studies, rectal doses were relatively high, which may be attributed to patient positioning and applicator size, potentially increasing the risk of rectal toxicity. they should ecplain that in the discussion.

Response 6: Indeed, the higher rectal doses are among the most curious findings in our results. We have contributed this result to certain factors, decribed in the following rows: 229-230, 237-238, and 241-244.

Comments 7: Limited Follow-Up Duration: Early toxicity was only observed over three months; longer follow-up could better assess late toxicity and long-term outcomes.

Response 7: We are grateful for your observation. In the conclusion of the revised manuscript, rows 264, 267-268, we added that long-term follow-up is still undergoing. Even though the results from it will provide more insight, we believe that early toxicity is still an important factor considering the increased radiobiological effectiveness of low-energy x-rays which may increase side effects, especially mucositis.

We are also thankful for pointing out the necessity for corrections in the grammar. The whole text has been examined and all mistakes found have been corrected.

Reviewer 2 Report

Comments and Suggestions for Authors
  1. Introduction: While the manuscript mentions that electronic brachytherapy requires less radiation shielding due to lower energy, other more reasons for choosing electronic brachytherapy should also be explained in detail to provide a comprehensive understanding of its advantages over traditional radioisotope-based methods.
  2. Active length: The term "active length" is mentioned in the manuscript but not defined. A clear explanation of what this term means is necessary.
  3. Treatment time: The prescription doses are consistently 12 Gy or 10 Gy with the same prescription point (5 mm outside the applicator). Why, then, is there a difference in total treatment time? An explanation is required in the discussion section.
  4. Plan validation: The "team members" involved in the joint decision-making process should be specified (e.g., how many and their roles). Additionally, the criteria for decision-making need to be defined. If no formal criteria exist, the decision-making process should be described in detail. Because this decision could be related to clinical outcomes.
  5. Table 3: Units of measurement are missing and should be included.
  6. Table 4: Are the values in this table averages? If so, the distribution (e.g., range, standard deviation, …) should also be provided for a more comprehensive understanding.
  7. Figure 3: More detailed explanations are needed for Figure 3. For instance, clarify what each contour represents, identify which part is the applicator, and describe the meaning of labels such as "01, 02, ... 06."
  8. The description of the patient's position during the procedure requires clarification. Specifically, does "gynecological position" refer to the lithotomy position? While the manuscript mentions the use of a vacuum bag, it is unclear whether the patient was in the lithotomy position during its use or if they were simply supine. Additionally, during treatment, it is stated that the legs are rested straight—was the vacuum bag designed for the lithotomy position still utilized in this case, or was a different setup employed? This should be explicitly detailed for better understanding.
  9. Decimal notation: The decimal notation in the manuscript alternates between periods and commas. It is necessary to standardize this notation for consistency.

By addressing these points, the manuscript will become more robust and informative, making it a valuable resource in the field.

Author Response

Thank you very much for taking the time to review this manuscript. Please find the detailed responses below and in the revised manuscript.

Comments 1: Introduction: While the manuscript mentions that electronic brachytherapy requires less radiation shielding due to lower energy, other more reasons for choosing electronic brachytherapy should also be explained in detail to provide a comprehensive understanding of its advantages over traditional radioisotope-based methods.

Response 1: Thank you for the valuable advice. We have added in the revised manuscript additional information in rows 60-63, as well as a new reference number 8. The method is mostly valued from an ecological point of view apart from its clinical results which are still being studied.

Comments 2: Active length: The term "active length" is mentioned in the manuscript but not defined. A clear explanation of what this term means is necessary.

Response 2: We have considered your suggestion and made the necessary correction by adding an explanation in rows 119-120.

Comments 3: Treatment time: The prescription doses are consistently 12 Gy or 10 Gy with the same prescription point (5 mm outside the applicator). Why, then, is there a difference in total treatment time? An explanation is required in the discussion section.

Response 3: Thank you for pointing this out. We have described in rows 222-224 of the revised manuscript that the treatment times vary depending on the dwell positions, which are calculated individually according to the proximity of organs at risk.

Comments 4: Plan validation: The "team members" involved in the joint decision-making process should be specified (e.g., how many and their roles). Additionally, the criteria for decision-making need to be defined. If no formal criteria exist, the decision-making process should be described in detail. Because this decision could be related to clinical outcomes.

Response 4: You are absolutely correct that the criteria for decision making must be mentioned . In rows 121-124 and the newly created Table 1, we have provided more information.

Comments 5: Table 3: Units of measurement are missing and should be included.

Response 5: Due to the addition of Table 1, it is now Table 4 and we have made the suggested correction.

Comments 6: Table 4: Are the values in this table averages? If so, the distribution (e.g., range, standard deviation, …) should also be provided for a more comprehensive understanding

Response 6: We are very thankful for this question and it served as a reminder to add all the missing data. The table is now numered as Table 5.

Comments 7: Figure 3: More detailed explanations are needed for Figure 3. For instance, clarify what each contour represents, identify which part is the applicator, and describe the meaning of labels such as "01, 02, ... 06."

Response 7: We have made the effort to add more details in the explanation. They can be found in rows 192-196.

Comments 8: The description of the patient's position during the procedure requires clarification. Specifically, does "gynecological position" refer to the lithotomy position? While the manuscript mentions the use of a vacuum bag, it is unclear whether the patient was in the lithotomy position during its use or if they were simply supine. Additionally, during treatment, it is stated that the legs are rested straight—was the vacuum bag designed for the lithotomy position still utilized in this case, or was a different setup employed? This should be explicitly detailed for better understanding

Response 8: We are thankful for pointing out the need for clarification here. The whole paragraph has been rewritten to make the position of the patient more clear. In rows 101-107 you can find the changes made.

Comments 9: Decimal notation: The decimal notation in the manuscript alternates between periods and commas. It is necessary to standardize this notation for consistency.

Response 9: We have standatized the notation in the revised manuscript.